# Cost-Effectiveness Analysis of Personalized Hypertension Prevention

**DOI:** 10.3390/jpm13061001

**Published:** 2023-06-15

**Authors:** Sen-Te Wang, Ting-Yu Lin, Tony Hsiu-Hsi Chen, Sam Li-Sheng Chen, Jean Ching-Yuan Fann

**Affiliations:** 1Department of Family Medicine, School of Medicine, College of Medicine, Taipei Medical University, Taipei 11031, Taiwan; wangader@gmail.com; 2Department of Family Medicine, Taipei Medical University Hospital, Taipei 10301, Taiwan; 3Institute of Epidemiology and Preventive Medicine, College of Public Health, National Taiwan University, Taipei 10663, Taiwan; happy82526@gmail.com (T.-Y.L.); chenlin@ntu.edu.tw (T.H.-H.C.); 4School of Oral Hygiene, College of Oral Medicine, Taipei Medical University, Taipei 11031, Taiwan; 5Department of Health Industry Management, School of Healthcare Management, Kainan University, Tao-Yuan 33857, Taiwan

**Keywords:** cost-effectiveness analysis, hypertension, personalized prevention

## Abstract

Background: While a population-wide strategy involving lifestyle changes and a high-risk strategy involving pharmacological interventions have been described, the recently proposed personalized medicine approach combining both strategies for the prevention of hypertension has increasingly gained attention. However, a cost-effectiveness analysis has been hardly addressed. This study was set out to build a Markov analytical decision model with a variety of prevention strategies in order to conduct an economic analysis for tailored preventative methods. Methods: The Markov decision model was used to perform an economic analysis of four preventative strategies: usual care, a population-based universal approach, a population-based high-risk approach, and a personalized strategy. In all decisions, the cohort in each prevention method was tracked throughout time to clarify the four-state model-based natural history of hypertension. Utilizing the Monte Carlo simulation, a probabilistic cost-effectiveness analysis was carried out. The incremental cost-effectiveness ratio was calculated to estimate the additional cost to save an additional life year. Results: The incremental cost-effectiveness ratios (ICER) for the personalized preventive strategy versus those for standard care were -USD 3317 per QALY gained, whereas they were, respectively, USD 120,781 and USD 53,223 per Quality-Adjusted Life Year (QALY) gained for the population-wide universal approach and the population-based high-risk approach. When the ceiling ratio of willingness to pay was USD 300,000, the probability of being cost-effective reached 74% for the universal approach and was almost certain for the personalized preventive strategy. The equivalent analysis for the personalized strategy against a general plan showed that the former was still cost-effective. Conclusions: To support a health economic decision model for the financial evaluation of hypertension preventative measures, a personalized four-state natural history of hypertension model was created. The personalized preventive treatment appeared more cost-effective than population-based conventional care. These findings are extremely valuable for making hypertension-based health decisions based on precise preventive medication.

## 1. Introduction

Since 1990, the polypill approach has been the primary method of hypertension prevention. Since 2003, the introduction of lifestyle changes has been the primary method of prevention, with a focus on pre-hypertension, a precursor phase to hypertension that JNC 7 proposed. The possibility of treating hypertension beyond what can be expected from the clinical management of stage I and stage II hypertension was raised. The personalized study of the natural history of hypertension has been made possible by the recently proposed precision medicine, and also considers the possibility of natural regression from pre-hypertension. Predicting the risk of a dynamic process of multi-state transition from a normal state to stage II hypertension through pre-hypertension and stage I hypertension, a previous study also developed a multi-state model with the incorporation of state-specific covariates [1]. Based on these risk scores, a population’s underlying hypertension risk can be stratified for the purpose of guiding a personalized preventive strategy. The epidemiological profiles of prevalence, incidence, and mortality, as well as the disparity of the disease natural history varying with gender before and after menopause, are just a few examples from previous studies [2,3]. Before conducting an economic appraisal, it is important to take into account age and gender differences. Younger women present a higher incremental cost-effectiveness ratio (ICER) compared to other age–gender-specific groups, according to Moran et al. [4].

Two common approaches to preventing hypertension are the population-wide approach and the high-risk approach. The former relates to the use of intervention strategies, such as lifestyle modifications, in the underlying general population on the basis of Rose theory [5], and the latter is a traditional method for choosing a high-risk group for receiving interventions, such as lifestyle modifications and prophylactic pharmacological interventions. Previous studies on the cost-effectiveness analysis of population-based hypertension control obtained similar results [6,7]. Community-based programs focusing on health behavior modifications and medication adherence were deemed cost-effective and may lower the long-term healthcare expenses [6]. According to a systematic review, a population-wide salt reduction may be cost-effective in hypertension prevention [7]. Other studies evaluated the cost-effectiveness of hypertension screening strategies for population-based interventions. Most strategies for hypertension screening were found to be cost-effective to prevent cardiovascular disease [8,9]. In the high-risk approach, polypill intervention was cost-effective [10] and even cost-saving in a Thai study [11]. Most hypertension interventions including pharmacotherapy only or pharmacotherapy plus health education or lifestyle changes were cost-effective in population studies in low–middle-income countries [12,13]. The economic evaluation of the results of cost-effectiveness analyses based on population-wide or high-risk approaches and on individualized approaches is therefore interesting. On the basis of the available evidence from the economic evaluation, both individual and population-based strategies would be suggested to achieve the greatest health gains at the lowest possible cost. [14]. Additionally, the cost-effectiveness of hypertension prevention strategies may be impacted by the natural regression from pre-hypertension to the normal state; however, this issue has not been previously taken into account when conducting an economic appraisal.

The most intriguing aspect of personalized preventive strategies is their economic evaluation in light of risk stratification based on individual risk scores, which runs parallel to the personalized disease natural history. A personalized preventive strategies’ cost-effectiveness analysis has never been carried out. Therefore, our goal was to conduct an economic analysis of a customized hypertension prevention strategy.

## 2. Materials and Methods

### 2.1. Study Framework

The framework for a systematic economic analysis for hypertension prevention is shown in Figure 1. The steps of creating a hypertension status, determining the state-specific risk for hypertension evolution, administering intervention strategies, such as those based on a population-wide universal approach, a population-based high-risk approach, and a personalized approach, and evaluating their effectiveness and cost are all included in the proposed analysis.

Based on the JNC 2003 classification of hypertension into normal status, prehypertension, stage I hypertension, and stage II hypertension [15], a four-state model was created. For the preclinical disease status of prehypertension, progression and regression were considered. Regression after the entry into stage I hypertension is not probable. Based on data from the Keelung community-based integrated screening program, Tseng et al. assessed the evolution of hypertension between states in 2012 [1].

Intervention strategies including the population-wide universal approach, the population-based high-risk approach, and the personalized approach were evaluated for the prevention of hypertension. Considering the varying natural history of disease evolution in terms of the rates of progression and regression, the effectiveness of these intervention strategies was determined. The transition probability allows determining the shifting of the stages and the possible decrease in hypertension-related complications and fatalities. The effectiveness of the intervention strategies can be measured by comparing outcomes in terms of person-years gained and quality-adjusted person-years gained. The cost associated with the intervention strategies was then taken into account in order to perform a cost-utility analysis (CUA) economically.

### 2.2. State-Specific Risk Scores

Based on the estimated findings on state-specific clinical weight published by Tseng et al. in 2012 [1], disease state-based risk scores can be generated. The effect of described covariates on the state-specific risk scores causes the transition probability to depend on the kind of risk factors for a subject. The population can be divided into groups based on the net force leading to the occurrence of stage I hypertension as determined by the state-specific risk scores by incorporating individual level state-specific risk factors and using a four-state Markov model.

### 2.3. Markov Decision Model

Figure 2 shows the Markov decision model for the economic analysis. In all decisions, a cohort for each prevention strategy was used to follow the four-state model-based disease natural history of hypertension. With time, the disease progression through its various states changed. Following the prognosis of three complications—end-stage renal disease (ESRD), stroke, and acute myocardial infarction (AMI)—and death attributable to the three complications, the Markov cycle tree was drawn. After the interventions, there were a total of seven complication nodes, including ESRD, Stroke, AMI, ESRD and Stroke, ESRD and AMI, and all three complications. Here, we represented the yearly dynamic change in disease status using a Markov node and assigned the corresponding parameters for cost and efficacy.

We used a decision node (□) to indicate the selected decision and used a chance node (◯) to show all possible outcomes with different probabilities. The symbol of [+] in Figure 2 denotes the subsequent tree for all events. In a Markov tree, the four-state natural history of hypertension was classified in normal state, pre-hypertension, stage I, and stage II. The Markov node (
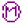
) was defined in the Markov cycle with a one-year follow-up for one cycle. The final states were denoted as terminal nodes (◁) after the simulation.

### 2.4. Cost-Utility Analysis

Based on risk groups and stage of hypertension, a cost-effectiveness analysis of a personalized strategy with varying levels of lifestyle modifications and awareness was conducted (Figure 3). Table 1 reports a list of the base-case parameters used to represent the progression of hypertension in the simulated cohort. The estimated results based on the empirical data of the Keelung prospective cohort study were used to derive the transition probabilities for the four-state Markov process, which included the three progress rates from the state of normal to prehypertension, from prehypertension to stage I hypertension, and from stage I hypertension to stage II hypertension, as well as one regression rate from prehypertension to normal [16]. In addition to the change in the hypertension states, prognostic factors were taken from the literature and included the likelihood of complications associated with each state of hypertension (blood pressure-specific hazard) taking into account three major end-organ damage events: myocardial infarction, stroke, and chronic renal failure. The annual rates for prehypertension, stage I hypertension, and stage II hypertension were specified as 0.0032, 0.0068, and 0.0095, respectively, in relation to the occurrence of myocardial infarction. The corresponding annual rates for the three hypertension states were 0.0008, 0.0018, and 0.0072, respectively, in relation to the Stroke complication. The annual rates for the three hypertension states in relation to the development of chronic renal failure were 0.002, 0.0321, and 0.0462. Because the severity of the disease affects how often complications occur, the difference in the annual rates for the occurrence of specific end-organ damage related to the states of hypertension reflects the insults caused by the spectrum of hypertensive disorder. Based on statistics from the literature, the mortality was also specified for each of the three complications.

It was also stated that the effectiveness of screening activities for lowering blood pressure depends on the stage of the hypertensive disorder. It is unlikely that screening activities and related awareness will lower blood pressure in stage II hypertension, which is a clinically evident disease. For the base-case, the compliance rates for the life modifications, screening activity, and medications were set at 60%, 70%, and 50%, respectively. Both direct and indirect costs were taken into account. The direct costs were determined based on either the Taiwanese Health Insurance program or out-of-pocket costs. The indirect costs included working day losses due to screening and production losses from illness to death. The discount rate was set at 3%.

The societal point of view was used. By dividing the marginal cost (e.g., the discount cost of universal care minus the discount cost of usual care) by the marginal effectiveness (e.g., the discount effectiveness of universal care minus the discount effectiveness of usual care), the incremental cost-utility ratio (ICUR) was calculated to estimate the additional cost to save an additional year of life. In order to perform a probabilistic cost-utility analysis, the Bayesian approach utilizing the Monte Carlo simulation was used. The simulated estimates of incremental cost, incremental utility, and ICUR were plotted and located in a four-quadrant cost-effectiveness plane using the Monte Carlo Simulation method for parameter uncertainty. All these analyses were performed by using TreeAge Pro 2019 software.

## 3. Results

Table 2 displays the findings of a cost-utility analysis (CUA) for the universal prevention of hypertension using three different approaches: a population-wide universal approach based on lifestyle modifications, a high-risk group approach, and a personalized approach, using stage I and stage II hypertension as the reference group.

The population-wide approach resulted in a gain of 0.0184 QALY for the entire population at an additional cost of USD 2226, resulting in an incremental cost-effectiveness ratio (ICER) of US 120,781 per QALY gained. The incremental cost ranged between USD 1000 and USD 3000, and the incremental effectiveness ranged between −0.06 and 0.16 according to the scatter incremental cost-effectiveness plot shown in Figure 4A. According to the acceptability curve, the two strategies had an equal chance of being cost-effective at a willingness-to-pay level of USD 180,000. When the level of WTP was greater than USD 300,000, the universal strategy was 74% likely to be financially advantageous (Figure 5A).

With an additional cost of USD 1346, the high-risk approach strategy increased QALYs by 0.0253, resulting in an incremental cost-effectiveness ratio (ICER) of USD 53,223 per QALY gained. According to Figure 4B, which shows the scatter incremental cost-effectiveness plot, the incremental cost ranged between USD 800 and USD 2000, and the incremental effectiveness ranged between -0.06 and 0.16. The acceptability curve demonstrates that the likelihood of both strategies being cost-effective at a willingness-to-pay level of USD 100,000 was equal. When the level of WTP was greater than USD 300,000, the cost-effectiveness of the high-risk approach strategy was 85% likely (Figure 5B).

Comparing the personalized strategy to standard care, the former was found to be more effective (+0.4772 QALY) and less expensive (USD −1583). Figure 4C shows the scatter incremental cost-effectiveness plot. The acceptability curve demonstrated that the likelihood of a personalized approach being cost-effective was virtually certain (Figure 5C).

## 4. Discussion

This paper provides two significant contributions. In terms of methodology, this is the first study to model the natural history of hypertension from normal state to stage II hypertension while taking into account natural regression from pre-hypertension to normal using the four-state disease natural history model. It is also the first study to show how to assess chronic diseases such as hypertension economically using population-wide, high-risk, and personalized preventive strategies that are part of universal preventive strategies. The second contribution consists in showing how to apply the suggested methodology to empirical data in community-based integrated screening data to produce empirical evidence for average-risk economic appraisal as well as risk-guided personalized economic appraisal. This is from the perspective of public and clinical applications. Below, is a brief discussion of both.

A randomized control trial provides the most reliable scientific evidence when examining the effects of any intervention on the management of hypertension. The alternative is to use modelling to estimate the disease natural history in the absence of intervention in order to create a hypothetical comparator group that is similar to the control group of a randomized controlled trial. This can be achieved when the precise prevention of cardiovascular disease considering hypertension natural history from occurrence to regression or progression with a randomized trial cannot be obtained [23]. One of the alternative strategies would be the digital twin design [24].

### 4.1. Methodological Aspects during the Economic Appraisal of Preventive Strategies for Hypertension

This study is the first to develop a method for evaluating the economics of both personalized and average-risk-based preventive strategies. Two common approaches—population-wide and targeted at high-risk groups—are used to prevent hypertension as well as its complications, deaths, and other undesired effects. One is wholly based on the Rose theory and shows that a change in blood pressure distribution can be attributed to the adoption of population-wide interventions such as dietary restrictions and increased physical activity. The other is a standard clinical management strategy that involves identifying the high-risk group with a cutoff above the blood pressure threshold and referring to this high-risk group for pharmacological intervention or clinical consultation and education. The gold standard for assessing the effectiveness of both strategies is to use a randomized controlled trial design. One of the most well-known examples of a population-wide strategy is provided by the DASH study and is based on dietary control and sodium intake monitoring [20,25]. A number of prophylactic pharmacological interventions for treating pre-hypertension and randomized controlled drug trials are useful examples [26,27,28]. Randomized controlled trials are the best method for determining whether a preventive strategy is effective and can also shed light on its mechanisms of action by determining whether it can reduce undesirable outcomes such as complications and fatalities. However, they necessitate extensive costs and follow-up logistics, as well as a longitudinal follow-up. As opposed to a high-risk group approach, a population-based randomized controlled study cannot be designed to assess the effectiveness of a population-wide approach, because the entire population has already been exposed to the intervention.

As an alternative, it is possible to model a disease natural history in the absence of treatment in order to develop a fictitious comparator group that resembles the control group in a randomized controlled trial. The outcomes of life years or quality-adjusted life years lost to related complications and death can be assessed in order to determine the efficacy and effectiveness of both preventive approaches. This can be carried out by observing how the stage distribution of hypertension can be changed by altering the age-specific average-risk-based disease natural history. The foundation of an economic appraisal of the average-risk-based cost-effectiveness (utility) analysis is the provision of cost information for each intervention as well as of the treatment costs for each state and its subsequent complications including death together with information on the effectiveness.

Age-specific average-risk-based natural history can be expanded to create a personalized-risk-based natural history model in order to further consider personalized preventive strategies. This model also incorporates a constellation of significant state-specific covariates to produce a risk score, which is then used to categorize the risk groups. Different risk groups are taken into consideration when assigning personalized preventive strategies. It is possible to conduct an individual economic appraisal for risk-guided preventive strategies.

### 4.2. Natural History of Hypertension for an Average-Risk-Based Population and a Personalized-Risk-Based Population

The construction of a disease natural history plays a crucial role in the execution of an economic appraisal because it is a fundamental requirement for providing a comparator for average-risk-based preventive strategies and risk-guided preventive strategies. Prior to 2003, the disease natural course was thought to start from a normal state, then advance to stage I, and finally advance to stage II. Pre-hypertension is a new included intermediate state between the normal state and stage I disease that JNC 7 proposed following the year 2003. The natural history of hypertension was built as follows: normal sate, pre-hypertension, stage I hypertension, stage II hypertension [15].

A personalized disease natural history model was created by incorporating state-specific covariates into a four-state Markov regression model. The risk score was used to categorize various risk groups. Additionally, dynamic curves specific to the progression and remission of hypertension were drawn.

### 4.3. Economical Aspecst of the Universal and Personalized Prevention Strategies for Hypertension

The primary contribution of this paper is the economic evaluation of both general strategies and personalized strategies. Given that the regression from pre-hypertension to normal state was taken into account in the present study, it makes sense that most ICUR values were higher than those obtained in the majority of earlier studies that did not account for the possibility of regression from pre-hypertension to normal state. It is very interesting to note that cost-saving personalized preventive strategies take into account various risk groups.

### 4.4. Limitations

There are three main limitations in this study. First, the efficacy of preventive strategies was not based on personalized efficacy because there is a lack of individualized evidence-based efficacy when personalized risk factors are taken into account. This requires a further study to provide the details on individualized efficacy. This is particularly important for distinguishing natural regression from the efficacy of an intervention. The second is that the costs of lifestyle modifications are very variable and depend on the underlying scenario in each country. This may also account for a higher ICER value noted in this study compared to previous ones. The third, we only considered three major complications that are associated with hypertension, that are, ESRD, stroke, and AMI, and other complications were omitted in this study, such as peripheral occlusive disease (PAOD), retinopathy, etc. We believe that the results of the economic evaluation would have indicated a higher cost-effectiveness if these complications were also considered in our analysis. In addition, the hypertension guidelines were revised by different medical organizations within the past several years. The major revisions included the adjustment of the BP staging, such as in the 2017 ACC/AHA hypertension guidelines [29], and of the blood pressure goal according to the age or comorbidities of the patients, such as in JNC 8 [30]. The inclusion of the above-mentioned elements would increase the complexity of the analytical model and will be carried out in future research based on the current study model.

## 5. Conclusions

To support a health economic decision model for the economic evaluation of both a universal and a personalized preventive strategy, a four-state natural history of hypertension model with a personalized four-state disease natural history was created. The main finding of this study was the cost-saving nature of the personalized preventive strategy. The reported findings are extremely valuable for making precise preventive medicine decisions for hypertension.

## Figures and Tables

**Figure 1 jpm-13-01001-f001:**
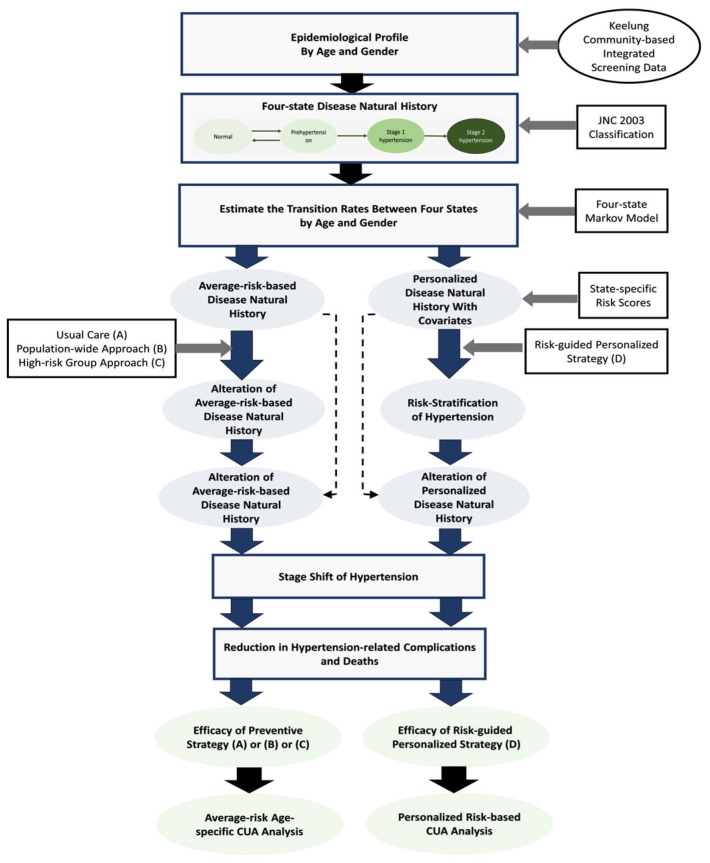
Study framework of the economic appraisal of the prevention of hypertension carried out in this work.

**Figure 2 jpm-13-01001-f002:**
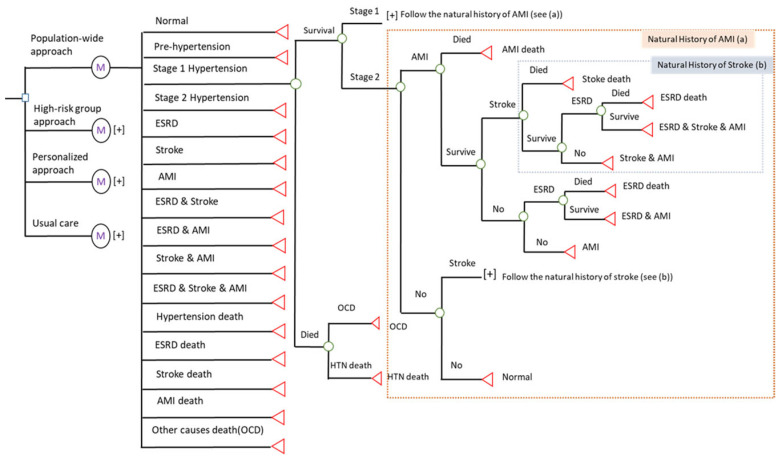
Markov decision tree for the cost-effectiveness of hypertension prevention. (**a**) The part of nature history of acute myocardial infarction. (**b**) The part of nature history of stroke.

**Figure 3 jpm-13-01001-f003:**
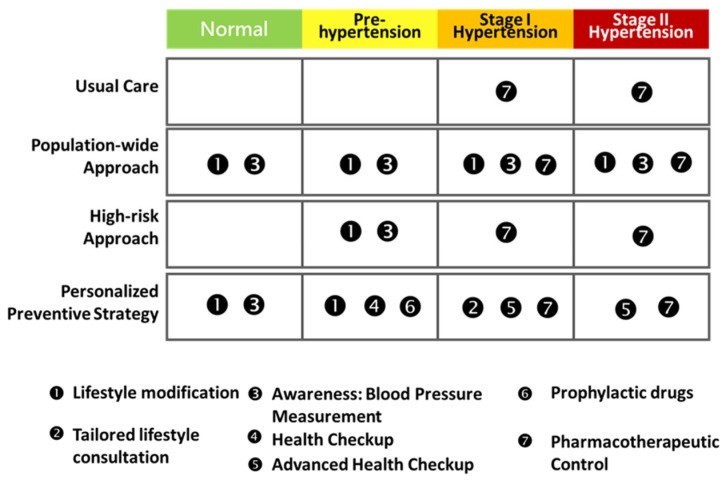
The four-stage hypertension progression with the four intervention strategies. The color indicates the disease risk from low to high.

**Figure 4 jpm-13-01001-f004:**
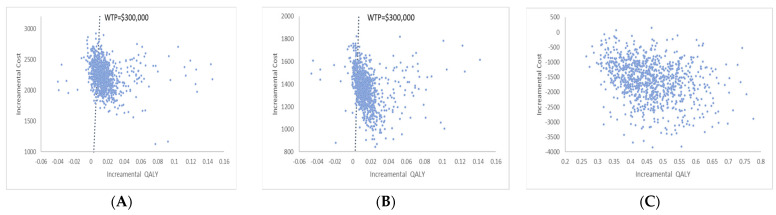
The scatter incremental cost effectiveness plot for hypertension prevention. (**A**) Population-wide approach vs. usual care (**B**) High-risk group approach vs- usual care (**C**) Personalized approach vs. usual care.

**Figure 5 jpm-13-01001-f005:**
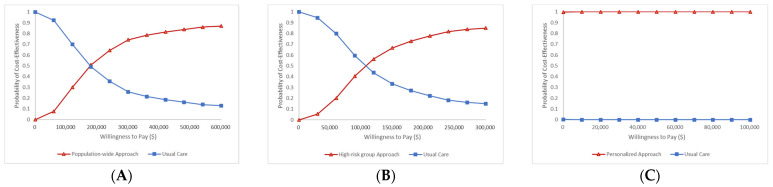
The acceptability curve for hypertension prevention. (**A**) Population-wide approach vs. usual care (**B**) High-risk group approach vs. usual care (**C**) Personalized approach vs. usual care.

**Table 1 jpm-13-01001-t001:** Base-case estimates and distributions for the cost-effectiveness analysis.

Variable	Base-Case Values	Distribution	Reference
Prevalence and disease natural history			
Prevalence of hypertension	Refer to Appendix A		Empirical data in Taiwanese community
Transition probabilities in subsequent years			Tseng et al., 2012 [1]
Specific hazard for blood pressure			Tseng et al., 2012 [1]
Myocardial infarction			Carlos et al., 2003 [17]; Hansson et al., 1999 [18]; Nelemans et al., 1998 [19]
Normal	0.003/yr	Beta (30, 9970)	
Prehypertension	0.003748/yr	Beta (32, 9968)	
Stage I	0.00633/yr	Beta (68, 9932)	
Stage II	0.008737/yr	Beta (95, 9905)	
Mortality, immediate	0.15	Beta (15, 85)	
Mortality, yearly	0.0311	Beta (311, 9689)	
Stroke			Carlos et al., 2003 [17]; Hansson et al., 1999 [18]; Nelemans et al., 1998 [19]
Normal	0.00075/yr	Beta (7.5, 9992.5)	
Prehypertension	0.000937/yr	Beta (9.37, 9990.63)	
Stage I	0.001675/yr	Beta (18, 9982)	
Stage II	0.00333/yr	Beta (72, 9928)	
Mortality, immediate	0.19	Beta (19, 81)	
Mortality, yearly	0.0201	Beta (201, 9799)	
ESRD			Carlos et al., 2003 [17]; Nelemans et al., 1998 [19]
Normal	0.000075/yr	Beta (0.75, 9999.25)	
Prehypertension	0.0000937/yr	Beta (20, 9980)	
Stage I	0.001213/yr	Beta (321, 9679)	
Stage II	0.001716/yr	Beta (462, 4538)	
Mortality	0.30		
Screening efficacy, lowering blood pressure			
For prehypertension	7/3 mm Hg		Sacks et al., 2001 [20]
For Stage I	16/-- mm Hg		
For Stage II	19.5/8.1 mm Hg		Insua et al., 1994 [21], Prospective Studies Collaboration, 2002 [22]
Compliance			
Life modification	60%	Beta (60, 40)	
Screening	70%	Beta (70, 30)	
Prophylactic medicine	50%	Beta (50, 50)	
Cost for myocardial infarction			
Cost of myocardial infarction per admission		Triangular (2706, 3006, 3068)	National Health Insurance Program, Taiwan
Cost of myocardial infarction per visit		Triangular (50, 55, 66)	
Cost for stroke			
Cost of stroke per admission		Triangular (1027, 1141, 1370)	
Cost of stoke per visit		Triangular (54, 59, 71)	
Cost for ESRD			
Annual Cost of ESRD (outpatients and administration included)		Triangular (18,000, 20,000, 24,000)	
Cost for hypertension			
Cost of hypertension per admission		Triangular (583, 647, 777)	
Cost of hypertension per visit		Triangular (45, 40, 54)	
Cost for life modification			
Cost for life modification low		Triangular (300, 450, 600)	
Cost for life modification mid		Triangular (300, 600, 900)	
Cost for advanced health check up		Triangular (300, 400, 500)	
Cost for life modification		Triangular (900, 1200, 1500)	
Cost for others			
Cost for screening/low awareness		Triangular (40, 80, 120)	
Cost for regular health checkup		Triangular (100, 200, 300)	
Cost of terminal care		Triangular (2000, 5000, 10,000)	
Cost of prophylactic drug		Triangular (60, 120, 180)	
Discount rate	3% (0–5%)	Triangular (0, 0.03, 0.05)	

**Table 2 jpm-13-01001-t002:** Cost-effectiveness analysis for hypertension prevention.

Strategy	Cost	QALY	Increment Cost	Increment QALY	ICUR
Usual care	16,455	13.4622			
Population-wide approach	18,681	13.4806	2226	0.0184	120,781
High-risk group approach	17,801	13.4875	1346	0.0253	53,223
Personalized approach	14,872	13.9394	−1583	0.4772	−3317

To both cost and QALY, an annual discount rate of 3% was applied. The time horizon was 20 years.

## Data Availability

The data analyzed during the current study are available from the corresponding author upon reasonable request.

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
