# Peer review of "Cost-Effectiveness Analysis of Personalized Hypertension Prevention"

_jpm, 2023, doi:10.3390/jpm13061001_

Round 1
Reviewer 1 Report
Dear Author/s,
I read your article carefully. Your article explores the cost-effectiveness of a personalized approach to preventing hypertension, combining lifestyle changes and pharmacological intervention. The study found that the personalized preventive strategy was more cost-effective compared to standard care. I hope your findings help policymakers to attention more to prevention than treatment.
Good Luck
Author Response
Thank you for your kind words and comments.
Reviewer 2 Report
The authors presented a research on the cost-effectiveness analysis of personalized hypertension prevention. No major concerns were noticed by the reviewer. Minor issues include:
1. Will the authors include more literature review on the cost-effectiveness analysis of hypertension prevention and other major diseases in the introduction.
2. Will the cost-effectiveness analysis results differ across different demographic characteristics and other epidemiological characteristics.
Author Response
Q(1) Will the authors include more literature review on the cost-effectiveness analysis of hypertension prevention and other major diseases in the introduction.
Ans: Thank you for your valuable comments. We have added more literature reviews on the cost-effectiveness of hypertension prevention studies. (Page 2, Line 14-29)
Q(2) Will the cost-effectiveness analysis results differ across different demographic characteristics and other epidemiological characteristics.
Ans: Thank you very much for your insightful remark. Our suggested tailored preventative strategy is based on risk stratification, which already takes into account demographic and other epidemiological characteristics. Therefore, we think these considerations may have an impact on the absolute values but won't have a big effect on the cost-effectiveness results.